# Event-Location Tracking in Narratives:
# A Case Study on Holocaust Testimonies

**Eitan Wagner**[†]    **Renana Keydar**[‡]    **Omri Abend**[†]

[†] Department of Computer Science    [‡] Faculty of Law and Digital Humanities

Hebrew University of Jerusalem

{first_name}.{last_name}@mail.huji.ac.il

## Abstract

This work focuses on the spatial dimension of narrative understanding and presents the task of event-location tracking in narrative texts, namely the extraction of the sequence of locations where the narrative is set. We present several architectures for the task that seek to model the global structure of the sequence, with varying levels of context awareness. We compare these methods to a number of strong baselines and ablated variants. We also develop methods for the generation of location embeddings and show that learning to predict a sequence of continuous embeddings is advantageous in terms of performance over predicting a string of locations. We focus on the test case of Holocaust survivor testimonies, motivated by the moral and historical importance of studying this dataset using computational means. The dataset further provides a unique case of a large set of narratives with a relatively restricted set of location trajectories. Our results show that models that are aware of the global context of the narrative can generate more accurate location chains. We corroborate the effectiveness of our methods by showing similar trends in an additional domain.[1]

## 1 Introduction

A primary goal of narrative analysis is to represent essential dimensions of stories in a schematic manner. One such essential dimension is the location or sequence of locations, in which the story takes place. In fact, location is such an important element in a story, that the ability to situate a story in a place is often viewed as one of the elements that distinguish narrative text from other types of texts (Piper and Bagga, 2022). Characterizing stories by their sequence of locations is also beneficial in that it provides a backbone for alignment between different stories–an important task in its own right (see, e.g., Ernst et al., 2022).

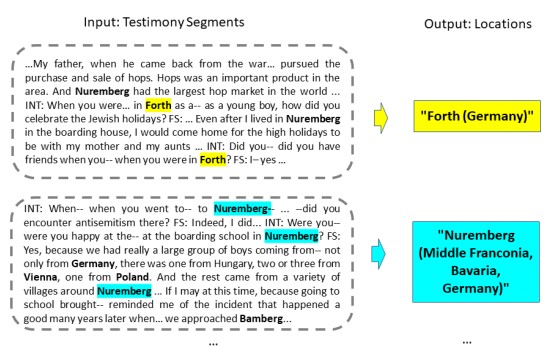

Figure 1: Event-Location Tracking in Holocaust testimonies. The output is the list of event locations which is not identical to the list of location mentions.

Nevertheless, and despite the wealth of NLP literature that has studied the expression of locations in texts (see §2), we are not aware of works that attempted to extract the trajectory or sequence of locations where a narrative story takes place.

In this work, we focus on transcriptions of Holocaust survivor testimonies, given in English. It is difficult to overstate the importance of this dataset for the study and memory of the Holocaust. With the inevitable perishing of the last living survivors, there is an imminent need to develop new modes of engagement with the vast amount of Holocaust testimonies stored in the archives. The application of NLP technology to the analysis of these testimonies has recently been strongly advocated (Artstein et al., 2016; Wagner et al., 2022). Indeed, NLP can aid in allowing researchers to gain insight from the entire collection of testimonies (tens of thousands of them), instead of focusing on small-scale, mostly manual, studies. We further argue that Holocaust testimonies have the additional, unique value for NLP, due to its combination of a large number of testimonies and a relatively restricted domain, in terms of themes and locations. This quality stands in contrast to that of typical narrative datasets (Sultana et al., 2022).

---

[1] Code is provided at https://github.com/eitanwagner/location-tracking

In the narrative framework, event-location tracking is unique in two ways. First, as in Gius and Vauth (2022), the focus is on *event* location, i.e., the location where the event takes place. For example, if a witness recounts events in the ghetto, and mentions someone who came from some town, then although the town's name is a mention of a location, it is not part of the event location. Second, the locations in narratives should be viewed as a *trajectory*. For example, migration from Europe to America is not merely a change of location since it follows a common post-war theme of Holocaust-related trajectories. The task of event-location tracking is therefore different from entity linking in its desired output – a sequence of segment-wise predictions for an entire document.

Trajectory extraction can be valuable for visualization and trajectory clustering (Bian et al., 2018). In addition, successful location extraction indicates aspects of long-range narrative understanding, a currently open problem in NLP (Yao et al., 2022).

In this work, we present various architectures for location tracking that vary in their degree of context awareness. In one architecture, we use a learnable transition matrix, thus taking into account the adjacent locations. In another, we design a hierarchical transformer that can attend to the whole document. We also design a method to generate embeddings for the locations. We compare our methods to a number of baselines that use strong language models but are limited in the context-length that can be taken into account.

We show that our models significantly outperform the baseline models. We find that, in general, models with larger context capabilities have higher performance in location tracking. We also find that the use of location embeddings, trained with additional metadata, can contribute to the performance. For validation, we construct an experiment in an additional domain and show that similar trends arise.

In conclusion, we find that the task of event-location tracking belongs, both conceptually and empirically, to the level of full narratives. We argue that this is an important step towards the understanding of narratives as full stories.

Our contributions in this work are as follows: (1) we present the task of event-location tracking for an entire narrative text; (2) we establish an experimental setup for the task, in the domain of Holocaust survivor testimonies; (3) we present a number of architectures for the task, capable of document-level processing; (4) we present a method for creating domain-dependent location embeddings, and show its value to the task at hand.

## 2 Previous Work

**Narrative Analysis and Segmentation.** Narrative schema analysis attempts to model the essence of event sequences. It is beneficial to extract a high-level sequential progression of a long story, giving an overview of the events and allowing alignment between relevant parts. For example, Antoniak et al. (2019) visualized the frequent topic paths in birth stories using segment-wise topic modeling.

To extract an interpretable sequential progression it is necessary to divide the long story into shorter segments. Zehe et al. (2021) formulated the task of dividing a long text into segments by scenes. Hotho et al. (2021) summarized a shared task for scene segmentation and concluded that the task remains challenging. Our work is somewhat similar to scene segmentation since location trajectories induce some type of segmentation and locations are a key component in scenes. Nevertheless, the definition of a scene involves other aspects, such as time and plot, making it harder to formalize a prediction task and apply domain knowledge to aid the task. We focus, therefore, on locations only, and seek to extract the trajectory of the whole narrative. For this purpose, we focus on coarse locations (such as cities) and not on detailed scenes (such as specific houses).

**Event Locations.** Some recent works have expressed the significance of event locations for narrative analysis. Piper et al. (2021) formulated a definition of narratives with event locations as a key part. Soni et al. (2023) formulated a task of grounding characters to locations. Other works extracted event locations for single events, such as events described in tweets (Kumar and Singh, 2019). However, no previous work, to the best of our knowledge, has studied the trajectory of locations in a full narrative.

**Toponym Resolution.** Toponym resolution describes the task of extracting and identifying location mentions in a text. This task can be seen as a subtask of Named Entity Linking, by considering only location ("GPE") entities. Therefore, toponym resolution can be carried out with a general-purpose entity linker. Cao et al. (2021) released a general-purpose entity linker based on autoregres-

sive language generation. As in most entity linkers, the entities are Wikidata items.

Nevertheless, it has been shown that special training for spatial entities has better performance, leveraging specialized location gazetteers such as Geonames [2], which is more detailed than Wikidata location items (Hu et al., 2023). Hu et al. (2022) presented a survey about location-specific entity extraction. Zhang and Bethard (2023) provided a system that first resolves the non-ambiguous location mentions and used them as context for the ambiguous ones. Wang and Hu (2019) provided an evaluation platform for toponym resolution.

**Location Embeddings.** Many methods exist for general-purpose text-span embeddings (Le and Mikolov, 2014; Reimers and Gurevych, 2019). Specifically for locations, Tian et al. (2022) generated location embeddings from human mobility trajectories using a graph-based approach. Kejriwal and Szekely (2017) generated location embeddings based on random walks over the Geonames location graph. Dassereto et al. (2020) proposed evaluation methods for location embeddings.

**Trajectory Modeling.** In contrast to toponym resolution, which focuses on short mentions, a different line of work seeks to extract document-level trajectories. Mathew et al. (2012) modeled human location trajectories by Hidden Markov Models (HMM). Sassi et al. (2019) replaced HMMs with convolutional neural networks applied on location embeddings. Lui et al. (2021) used LSTM-based models for pedestrian trajectory prediction.

In our work, we propose the extraction of location trajectories for events in texts. There are two key differences between this and the existing work, which focuses on trajectory modeling for lists of location coordinates. First, event locations may not be explicitly mentioned in the text, therefore requiring non-trivial mention extraction. Second, locations in texts represent more than simple coordinate values. For example, a concentration camp might be close to a city but they clearly represent different thematic locations.

## 3 Task Definition

Our setting is the following: given a text, divided into initial segments, $\mathbf{x} = x_1, x_2, ..., x_n$, we wish to predict a sequence of locations $\mathbf{y} = y_1, y_2, \ldots y_n$, such that each location $y_i$ corresponds to the text

[2] http://geonames.org/

segment $x_i$. The location sequence induces another segmentation that is based on locations. We work with a closed set of locations $\mathcal{Y}$, i.e., $\forall i. y_i \in \mathcal{Y}$.

It is instructive to compare this task to NER for locations. Whereas NER is a prediction task at the phrase level, our task's focus is on the document as a long sequence of events. This change of focus implies some unique properties:

- In location tracking, the locations are to be assigned to spans of multiple sentences, that describe a locally confined cluster of events, and not to individual phrases as in NER. We do not see much value in marking the exact token-level boundaries of event locations since they are not necessarily clear from the text. In our case, we make a practical choice and use units of multiple sentences, as this is the scope of the available labeled data.

- In location tracking, the prediction is at the document level, with possible dependencies throughout the entire document. This property requires strong long-context capabilities. For example, a segment describing a visit to the narrator's hometown might not explicitly mention the town's name. The location name should be predicted as the event-location based on earlier mentions or external knowledge.

- Whereas NER annotates mentions, event-location tracking often requires inferring the location implicitly. Indeed, the task requires assigning a location to events even if no explicit location is mentioned. Conversely, many locations are mentioned without acting as event locations.

We also note that, specifically for testimonies, we differentiate between the locations of described events, which the task addresses, and the locations of the event of giving the testimony (the ground), which are not part of the location trajectory.

## 4 Methods

Location tracking is a structured prediction task, as there are dependencies between the segment labels $y_i$. For tasks like this, the global context of the document, and not just the individual segments, can be helpful. To this end, we propose three different approaches with different degrees of context awareness.

### 4.1 Deep CRF

We design and implement a linear CRF model (Lafferty et al., 2001) with transformer inputs. We first

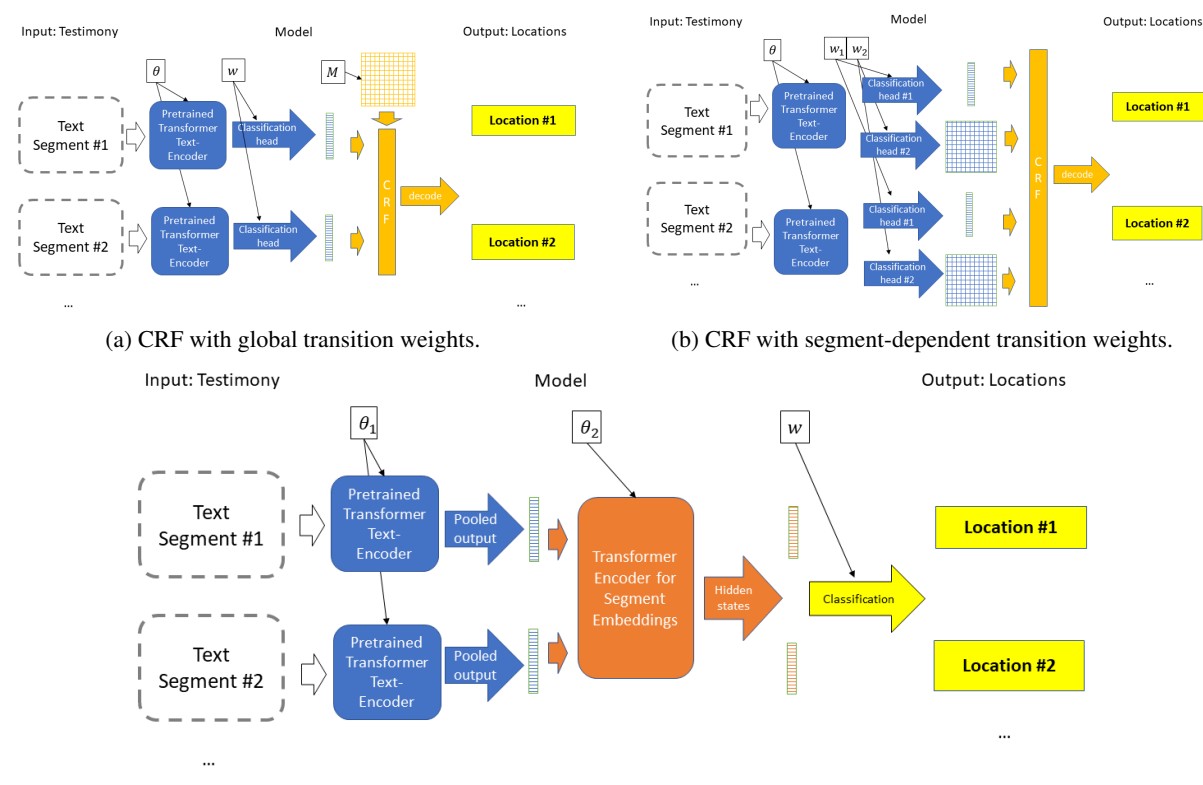

(a) CRF with global transition weights.

(b) CRF with segment-dependent transition weights.

(c) Hierarchical transformers.

Figure 2: Overview of the proposed models. Notation is as defined in Section 4.

finetune a transformer for segment location classification and use the output logits as inputs for the CRF module.[3] We propose two variants: the first is a more straightforward approach that uses a global transition potential matrix, while the second predicts the transition potentials "on the fly."

**Global Transition Probabilities.** In the first version, the model consists of one transition matrix that is learned during training. The transition matrix is a parameter of the CRF component. See figure 2a. We denote the global transition model with CRF-G. Formally we can write:

$$P_{\text{CRF-G}}(\mathbf{y}|\mathbf{x}) = \frac{\exp\left[\sum_{i=1}^{n}\left(E_{\theta}(x_i)\cdot w + M_{y_{i-1},y_i}\right)\right]}{Z(\mathbf{x};\theta,w,M)} \quad (1)$$

where $E_{\theta}$ is the transformer encoder, $w$ are the weights of the linear layer on top of the classifier, and $M$ is the transition matrix. $Z$ is the partition function. $\theta, w$ and $M$ are learnable parameters.

In this version, the transition matrix is a function of the entire training dataset and does not depend on the specific local segment. This assumption has

---

[3]Our implementation for the CRF model is based on the TorchCRF package — https://github.com/s14t284/TorchCRF — with minor modifications.

limitations. For example, in most cases, the probability of staying in the previous location is highest, but if the previous segment describes migration or deportation then the probability of staying should be low. For this reason, we present another version with local probabilities.

**Local Transition Probabilities.** In the second version, the transition matrix is the output of a transformer that receives the segments as inputs and outputs a transition matrix. This allows the use of different transition probabilities based on the content of the current segment. The architecture of the transition estimator is similar to the classifier, but the output is a (flattened) logit matrix instead of a logit vector. Each entry in the matrix represents the potential of transitioning between two location labels. In this version, all the learned parameters are part of the transformer component and its heads. See figure 2b. We denote the local transition model by CRF-L. Formally:

$$P_{\text{CRF-L}}(\mathbf{y}|\mathbf{x}) =$$
$$\frac{\exp\left[\sum_{i=1}^{n}\left(E_{\theta_1}(x_i)\cdot w_1 + E_{\theta_2}(x_{i-1})_{y_{i-1},y_i}\cdot w_2\right)\right]}{Z(\mathbf{x};\theta_1,\theta_2,w_1,w_2)} \quad (2)$$

where $E_{\theta_1}$ and $E_{\theta_2}$ are the transformer encoders; $E_{\theta_1}$ outputs a vector for a text span and $E_{\theta_2}$ outputs a matrix. $w_1$ and $w_2$ are the weight vectors of the linear layers on top of $E_{\theta_1}$ and $E_{\theta_2}$, respectively. $Z$ is the partition function.

**Learning.** In both versions, the segment-level classifier is first finetuned with locally labeled segments. Then the CRF optimization is performed with the negative log-likelihood loss of the sequence. As in CRF models, the likelihood includes an exponential number of summands but can be computed efficiently with dynamic programming. Gradient updates can either include only the transition component (a matrix for the global version or a classification head for the local version), with the local classifier frozen, or can include further finetuning at the segment-level classifier too.

## 4.2 Hierarchical Transformer

In a similar manner to the CRF model, we design a hierarchical transformer with one transformer that encodes the text of each segment, and a second transformer that encodes the sequence of segment encodings. See Figure 2c.

This architecture has two potential advantages over the CRF ones. The first is that a linear CRF attends to one previous location only, thus limiting the available context. Extending the context involves a high computation cost. The second advantage is the larger flexibility of the hidden states. As opposed to the CRF model, which outputs probabilities, the transformer component outputs raw hidden states that can be used either as inputs for a classification layer or as the actual representations given a set of predetermined embeddings.

Given the flexibility of this architecture, we design two versions. The first uses discrete locations, and the second uses location embeddings.

**Discrete Locations.** This model is constructed with two components. First, a segment-level transformer text encoder receives a text segment and outputs an embedding vector. Second, a transformer encoder model receives a sequence of embeddings and outputs embeddings for sequence classification. We denote this model with HITRF. Formally:

$$\forall i \in [n]:$$
$$P_{\text{HITRF}}(y_i|\mathbf{x}) = softmax\big(\big[E_{\theta_2}(E_{\theta_1}(x_j)_{j=1}^n)\cdot w\big]_i\big) \quad (3)$$

where $E_{\theta_1}$ and $E_{\theta_2}$ are the transformer encoders; $E_{\theta_1}$ outputs a vector for a text span and $E_{\theta_2}$ outputs

hidden states for each vector inputs. $w$ is the weight vector for the linear layer on top of $E_{\theta_2}$.

**Location Embeddings.** The previous models use a discrete list of locations. In this setting, classification is done in a "one-hot" setting. A better option is to leverage the properties of the locations by creating location embeddings. This allows the use of zero-shot capabilities of the models, as well as the use of similarity between locations.

The embeddings are obtained by a sentence embedding model trained on natural language inference (NLI) data, with additional domain-specific data derived from the thesaurus and the segment annotations.

Using embeddings does not require structural changes to the architecture. The only difference is that the classification layer (i.e., the layer before the softmax) is initialized with the location embeddings and remains frozen during training. By doing so, we effectively predict the locations by projecting the inner representations onto the embeddings. We denote this model by HITRF+EMB.

## 4.3 Baseline Models

For baselines, we present several methods that do not take the entire document into account. Predictions are made for each segment independently, taking into account very limited additional context. These methods are simple and efficient. We intend to show that simplicity comes with a significant fall in performance.

**Independent Classification.** This method extracts a location sequence as independent classification tasks. We train a transformer model to classify the location and use this to predict the segment location independently, i.e., without taking into account the relationship between predicted classes. This is the same as $E_\theta(x)\cdot w$ in the proposed models. We denote this model with INDEPENDENT.

**Greedy Decoding.** In this method, we train a transformer to predict a location for a segment given some previous locations and segments. [4]

For decoding, we predict a location given the first segment and two 'START' (one for an empty segment), and then recursively predict the location for a segment given the previous segment and

---

[4]In our experiments, we used the two preceding locations and one preceding segment. The input format was: $y_{i-2}$ [SEP] $x_{i-1}$ [SEP] $y_{i-1}$ [SEP] $x_i$, where $x$-s are text segments and $y$-s are locations. The model was trained to predict $y_i$.

two previously predicted locations. We denote this model with GREEDY. Another version of this method used beam search for decoding. The results were not significantly different.

In the independent and greedy baseline methods, we also divided the testimony into 10 bins and inserted the bin number of the segment as part of the text. This gives the classifier some additional information beyond the segment itself.

**Location Entity Selection.** In this method, we apply NER and predict the location based on the entity mentions. We use a pretrained zero-shot classifier to predict the location based on the concatenation of all mentions of the locations in the segment, obtained from a NER tool. We denote this model with ENTITY.

**ChatGPT.** Another baseline we experiment with is zero-shot classification with ChatGPT.[5] We give the model instructions to return a location category from the specified list. For each segment, we input the instructions and the previous segment and its response, with an exception for cases where the response was not a category on the list, in which case, we did not input the previous segment and response for the next segment.[6] After receiving a list of predictions, we ignored all responses that were not on the list and instead kept the previous location category that was on the list (or used START if there was no previous location). See appendix B for the full prompts.

# 5 Experimental Setup

## 5.1 Data

**Location Data.** Our main data consists of 1000 Holocaust survivor testimonies, received from the Shoah Foundation (SF).[7] All interviews were conducted orally by an interviewer, recorded on video, and transcribed as time-stamped text. The lengths of the testimonies range from 2609 to 88105 words, with a mean length of 23536 words.

Each testimony recording was divided into segments, typically a segment for each minute. Each segment was indexed with labels, possibly multiple. The labels are all taken from the SF thesaurus.[8] The thesaurus is highly detailed, containing $\sim 8000$ unique labels across the segments.

---

[5] https://chat.openai.com/
[6] This is done to avoid propagation of this type of error.
[7] https://sfi.usc.edu/
[8] https://sfi.usc.edu/content/keyword-thesaurus

For the purpose of location tracking, we used index labels that can be seen as describing locations, such as cities, ghettos, concentration camps, etc. We performed our experiment with the category name (e.g., "Ghettos in Poland") and not the exact location name (e.g., "Warsaw (Warsaw, Poland: Ghetto)"). The conversion to categories was based on the hierarchy in the SF thesaurus, with minimal adaptation, that was done by Holocaust research experts. We ended up with 105 categories for locations and two more labels for 'START' and 'END'. The label set has 55 categories for cities (including moshavim and kibbutzim), 13 for concentration camps, 11 for administrative units, 9 for ghettos, 9 for DP camps, 9 for refugee camps, and one category for death camps. The full list can be found in appendix A.

Using location categories allows us to significantly reduce the number of categories (we have over 1500 exact locations). Reducing the number of categories inevitably results in some loss of information, but the categories still provide rich information useful for many tasks such as location-based segmentation, alignment between testimonies, and creation of high-level summaries.

We filtered out testimonies that did not follow the same labeling as the others, such as those with longer segments or very few location labels. We clarify that we did not discard any parts of the testimony text. For segments without location labels, we assumed the previous location is maintained, An exception is a case of a segment that was labeled as including a visit, in which case the next segment assumed the location before the visit. We ended up with 585 testimonies labeled with locations for each 1-minute segment. As the described process includes some heuristic labeling, which inevitably introduces noise, we manually proofed the labeling of a random set of 53 testimonies. These testimonies were held out for all steps of training, including embedding training and hyperparameter tuning, and were used only as a test set. Out of the remaining testimonies, we used 90% for training and 10% for validation.

## 5.2 Model Specifics

**General Considerations.** In all models that involved training, text segments were encoded with a pretrained transformer model. We decided to use the LUKE model (Yamada et al., 2020) since it was explicitly trained for entity-aware tasks. We used

the pretrained luke-base from Huggingface hub.[9] The outputs of the encoder were pooled using the default HuggingFace-transformer pooler. Specifically, the LUKE-pooler applies a linear transformation and *tanh* activation to the embedding of a special token that is placed at the beginning of the sequence.

**Baseline Models' Specifics** For the independent and greedy baselines, we finetuned LUKE for location classification with the standard Huggingface Trainer settings. For independent classification, the input was only the segment text and the relative location in the testimony, and for the greedy decoder, the input included the previous location-category label. For beam search, we used $k = 10$ beams.

For the entity selection baseline, we used the Holocaust NER model for entity recognition,[10] and bart-large[11] for zero-shot classification.

**Initialization.** The CRF transition matrix can be initialized either randomly or manually. Our experiments showed that initializing the prior transition matrix as a (smoothed) diagonal matrix yields the best results and this is what we used.

In HITRF, for the transformer for segment embeddings, we used a randomly initialized DeBERTa (He et al., 2021) architecture.

**Intermediate Training.** In the trained models, besides the pretraining for the first transformer, we added an intermediate step of training at the segment level. In all architectures, the segment-level text-transformer components (both classification and transition ones) were trained with the standard Huggingface trainer, with *batch-size*=4, *learning-rate*=$5 \cdot 10^{-5}$, for 3 epochs. This was done also in the hierarchical transformer architectures, with the only difference being that the classification head was used for this step only and no further.

**Finetuning.** The main training step is finetuning with whole documents. In one version this finetuning updates only the weights of the last component ($\theta_2$) – the CRF in the first architecture and the second transformer in the others. This method is fast and uses less memory, but it does not enable the usage of global properties on the segment level. In another version, we update all weights (i.e., also

$\theta_1$). This type of training requires some form of gradient accumulation for memory efficiency. [12]

In our experiments, we applied full gradient updates to all models besides CRF-L. This is due to technical complexity and the fact that the hierarchical transformer models yield better performance.

For all architectures, we used *batch-size*=1. We trained CRF-L, CRF-G and HITRF+EMB for 20 epochs, and HITRF 10 epochs. For the segment-level text-transformer, in the full gradient setting, we used *learning-rate*=$5 \cdot 10^{-6}$. For the CRF matrix, we used *learning-rate*=$5 \cdot 10^{-5}$ and *weight-decay*=$10^{-6}$. For the transformer with segment embeddings, we used *learning-rate*=$10^{-5}$.

**Location Embeddings.** Using the sentence-transformer package,[13] we finetuned LUKE for sentence embeddings. First, we finetuned with the combination of the SNLI (Bowman et al., 2015) and MultiNLI (Williams et al., 2018) datasets and with the Multiple Negative Ranking loss.[14] We added domain-specific training examples and further finetuned with them. The extra examples were testimony segments with the SF location labels and location terms from the SF Thesaurus with their corresponding descriptions. We note that even though the predicted output is on the category level, the location embeddings were trained also for the exact location level (i.e., for each city name). The pairs were converted to NLI form with standard prompts. See appendix C for further details on the embedding training.

### 5.3 Evaluation Methods

**Location Chains.** Our models output a sequence of locations for a predefined list. For pointwise comparison, we only consider exact matches and

---

[12]In the forward pass, $E_{\theta_1}$ receives segments and outputs corresponding (pooled) encoding vectors, and $E_{\theta_2}$ receives all these vectors and outputs a sequence of locations. The loss is calculated by the outputs of $E_{\theta_2}$, which are determined only after all segments were encoded (i.e., the input segments cannot be divided into batches). Encoding all segments before updating requires many copies of the computational graph, which requires a large amount of GPU memory. To allow batching, we do the following: We run a forward pass for $E_{\theta_1}$ without gradients, encoding each segment. With these encodings, we run a forward pass for $E_{\theta_2}$, with gradients, compute the loss, and backpropagate through $E_{\theta_2}$, obtaining (partial) gradients. We run forward again through $E_{\theta_1}$, this time with gradients, and backpropagate with the previously obtained gradients. The last forward pass can be done in batches.

[13]https://www.sbert.net/index.html

[14]https://www.sbert.net/docs/package_reference/losses.html#multiplenegativesrankingloss

[9]https://huggingface.co/studio-ousia/luke-base

[10]https://ner.pythonhumanities.com/intro.html

[11]facebook/bart-large-mnli

ignore the different levels of similarity between different locations. Although it is possible to take similarity into account, for example, by using embeddings, this will introduce bias to the evaluation.

One measure we used was Python's difflib SequenceMatcher (SM) score, which is based on the *gestalt pattern matching* metric (Ratcliff and Metzener, 1988). This metric sums the longest common substrings in a recursive manner, and divides by the total length, attempting to reflect human impression for similarity. In this metric, a higher score means stronger similarity.

Another sequence measure we used is the Damerau–Levenshtein edit distance (Edit, Damerau 1964). This measure defines the distance between two sequences as the minimal number of insertions, deletions, substitutions, or transpositions in order to get from one sequence to the other. Since the number of edits depends on the number of elements in the sequence, we normalized the distance by the number of locations in the reference document.[15] For the Edit distance, lower is better.

For completeness, we also report the element-wise accuracy (ACC) of the predictions.

## 6 Results

Scores for location tracking in Holocaust testimonies are in Table 1. We see that the proposed models significantly outperform the baseline models, with the best baseline (GREEDY) showing similar performance to the lowest performing proposed model (CRF-G). Regarding the CRF models, we can see that using locally predicted transition matrices can improve performance. The hierarchical transformer models show improvements, with the best results for the location transformer model with location embeddings.

We report one score for the greedy algorithm as we find that adding beam search had very little effect. For comparison, we mention here that fixed prediction for the most common category (which is "cities in Germany") gave much worse results: Edit = 0.84, SM = 0.07, and Accuracy = 0.11.

## 7 Discussion

Our experiments show that document-level architectures are significantly better at event-location

---

[15]The normalized distance might be larger than 1 if the predicted number of topics is larger than the real number. This normalization is commonly known in the literature as *word error rate*.

| Model | Edit | SM | ACC |
|---|---|---|---|
| INDEPENDENT | 0.64 | 0.19 | 0.36 |
| GREEDY | 0.58 | 0.32 | 0.42 |
| ENTITY | 0.7 | 0.17 | 0.28 |
| CHATGPT | 0.72 | 0.19 | 0.27 |
| CRF-G | 0.57 | 0.27 | 0.42 |
| CRF-L | 0.53 | 0.38 | 0.47 |
| HITRF | 0.4 | 0.43 | 0.59 |
| HITRF+EMB | **0.37** | **0.49** | **0.63** |

Table 1: Performance of the various models for event-location tracking in Holocaust testimonies, in terms of Sequence Matching (SM), Edit distance, and Accuracy (ACC). For SM and ACC higher is better and for Edit lower is better. The models in the top part are the baseline models. CRF-G and CRF-L are the global and local versions of the proposed CRF architecture, and HITRF and HITRF+EMB are the proposed hierarchical transformer architectures without and with location embeddings. All models except CRF-L were trained with full gradient updates.

tracking compared to local methods. This includes both fine-tuned transformers and strong zero-shot models, such as ChatGPT, and it is also true for such models as LUKE that are specifically designed to be aware of named entities.

Among the baselines, we see that simple local classification obtains poor results. This indicates that the task benefits from context. The zero-shot models seem to struggle to comply with the strict prediction format. This might be due to the abundance of implicit locations. The greedy model shows relatively good results, but its inferiority compared to the global models shows the importance of taking into account a larger context. Greedy's rough similarity in performance to the CRF-G model shows that the greedy models can balance the local signal (i.e., prediction based on the current text segment) and the transition signal (i.e., accounting for the previous location), without requiring global normalization.

Among our models, we see that the hierarchical transformer models perform better than the CRF models. This might be due to the larger context for the second level and the ability to capture more information in the first-level embeddings. In general, the experiment results follow a pattern in which models that are more expressive in terms of global context have better performance. Hierarchical transformer models are also robust to large sets of locations, in terms of memory, since the

embedding size is fixed. This is opposed to the CRF models, for which the transition matrix grows quadratically with the number of locations.

We also see that using specialized location embeddings improves performance over the test set. The location embeddings are also important in order to allow better adaptation to other sets of locations, such as more fine-grained categories, or even the exact location names, using the zero-shot capabilities of the embedding models to compensate for sparsity of data.

Manual inspection of the HITRF outputs, shows that in most cases the predicted trajectory is very similar to the gold one. The cases in which the performance was low seem to involve origin countries that are less frequent in the testimonies (like Lithuania or Luxembourg).

We also consider metrics that asses the models' performance in predicting location changes, while ignoring the exact location. For this, we use the F1-score for a binary prediction of whether a location change is made after a given segment. Interestingly, we find that with this metric, the greedy model (F1 = 0.29) and CRF-L (F1 = 0.24) outperform the HITRF models (F1 $\approx$ 0.17). CRF-G (F1 = 0.16) and INDEPENDENT (F1 = 0.1) under-perform the HITRF models. This result implies that the predicted trajectories, while fairly accurate, are not reliable enough as a means of segmentation. That is, it seems that the HITRF models succeed in capturing an overview of the testimonies, but struggle with exact boundaries, while models with local-transition components are more accurate in this regard.

In order to learn about the impact of bidirectionally on performance (separately from the impact of structured prediction), we designed an additional oracle baseline, based on an ensemble of greedy models. We trained a greedy model to predict a location given the following locations and segments (in a similar manner to the greedy baseline). Its results were slightly inferior to the original greedy model. We then constructed an oracle baseline in which we predict sequences in both directions and choose the backward predicted location if it is the correct one. The results were Edit = 0.41, SM = 0.45 and Accuracy = 0.59. These results are similar to HITRF but still not as good as HITRF+EMB. despite the oracle information available to this method. These results then show that even under idealized conditions, a simple bidi-

rectional model is not as strong as the best models presented here, underscoring the importance of taking into account a larger context.

An interesting feature of the CRF models is the ability to extract matrices with first-order transition scores. In CRF-G, we have a single matrix for the entire set of testimonies. In CRF-L, we have a function that assigns a unique matrix for a given segment. Although the matrix depends on the segment, our inspection shows that the generated matrices tend to be similar, allowing us to observe both general and segment-specific trends. In appendix E we present examples of matrices and discuss the trends observed in them.

Although our experiments focus on the test case of survivor testimonies, the techniques developed here are applicable more broadly. To validate this claim, we report in Appendix D results for an additional test domain. The results show a similar trend, thereby lending support to our claim as to the importance of viewing this task as a global (document-level) task.

## 8 Conclusion

We presented the task of event-location tracking in narrative texts and proposed various methods for tackling it.

We showed that our methods significantly outperform several baseline methods in terms of the trajectory of locations that the model generates. Between architectures, we showed that additional context improves results. We also showed that using location embeddings improves results, in addition to their importance for possible generalization. We corroborate our findings in another domain.

In addition to the technical contribution of this work, it also makes important first steps in analyzing spoken testimonies in a systematic, yet ethical, manner. As such, our work contributes to the domain of Holocaust and memory studies, designing new methods of browsing and analysis that augment the current efforts of scholars and practitioners alike. More broadly, this work makes a significant contribution to the growing field of computational literary studies by offering new models of narrative analysis, based on event locations.

### Limitations

Our experiments involve a specific and unique domain. While we did test on an additional domain, the main data and conceptual framework, such as

the location categories, are limited in this respect. Nevertheless, the domain provided an important well-motivated test case also from the NLP perspective.

## Ethical Considerations

We abided by the instructions provided by each of the archives. We note that the witnesses identified themselves by name, and so the testimonies are not anonymous. Still, we do not present in the analysis here any details that may disclose the identity of the witnesses. We intend to release our codebase and scripts, but those will not include any of the data received from the archives; the data and trained models used in this work will not be given to a third party without the consent of the relevant archives. The testimonies can be made accessible for browsing and research by requesting permission from the SF archive.

## Acknowledgements

The authors acknowledge the USC Shoah Foundation - The Institute for Visual History and Education for its support of this research. We thank Prof. Gal Elidan, Dr. Gabriel Stanovsky, and Noam Weis for their valuable insights, and Nicole Gruber, Yelena Lizuk, Maayan Aharoni, Inbal Neumark, and Asaf Gershon for their research assistance. This research was supported by grants from the Israeli Ministry of Science and Technology, the Israeli Council for Higher Education, and the Alfred Landecker Foundation.

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

# A  List of Location Categories

1. START

2. END

3. German concentration camps in Austria

4. German concentration camps in Belgium

5. German concentration camps in Czechoslovakia

6. German concentration camps in Danzig (FC)

7. German concentration camps in Estonia

8. German concentration camps in France

9. German concentration camps in Germany

10. German concentration camps in Latvia

11. German concentration camps in Poland

12. German concentration camps in the Netherlands

13. German concentration camps in the USSR

14. German death camps in Poland

15. German prisoner of war camps in France

16. German prisoner of war camps in Germany

17. Slovakian concentration camps in Czechoslovakia

18. Soviet concentration camps in the USSR

19. administrative units in Belgium

20. administrative units in Canada

21. administrative units in Denmark

22. administrative units in Hungary

23. administrative units in Israel

24. administrative units in Italy

25. administrative units in Lithuania

26. administrative units in Romania

27. administrative units in Spain

28. administrative units in the Netherlands

29. administrative units in the United Kingdom

30. cities in Argentina

31. cities in Australia

32. cities in Austria

33. cities in Belgium

34. cities in Bulgaria

35. cities in Canada

36. cities in Chile

37. cities in China

38. cities in Colombia

39. cities in Cuba

40. cities in Cyprus

41. cities in Czechoslovakia

42. cities in Danzig (FC)

43. cities in Denmark

44. cities in Dominican Republic

45. cities in Ecuador

46. cities in Egypt

47. cities in Estonia

48. cities in Finland

49. cities in France

50. cities in Germany

51. cities in Ghana

52. cities in Greece

53. cities in Hungary

54. cities in Iran

55. cities in Iraq

56. cities in Ireland

57. cities in Israel

58. cities in Italy

59. cities in Japan

60. cities in Latvia

61. cities in Libya

62. cities in Lithuania

63. cities in Luxembourg

64. cities in Mexico

65. cities in Morocco

66. cities in New Zealand

67. cities in Nigeria

68. cities in Norway

69. cities in Paraguay

70. cities in Poland

71. cities in Portugal

72. cities in Romania

73. cities in South Africa

74. cities in Spain

75. cities in Sweden

76. cities in Switzerland

77. cities in Uruguay

78. cities in Yugoslavia

79. cities in the Netherlands

80. cities in the USSR

81. cities in the United Kingdom

82. cities in the United States

83. displaced persons camps or installations in Austria

84. displaced persons camps or installations in Germany

85. displaced persons camps or installations in Germany: British zone

86. displaced persons camps or installations in Germany: US zone

87. displaced persons camps or installations in Italy

88. ghettos in China

89. ghettos in Czechoslovakia

90. ghettos in Greece

91. ghettos in Hungary

92. ghettos in Latvia

93. ghettos in Poland

94. ghettos in Romania

95. ghettos in the Netherlands

96. ghettos in the USSR

97. kibbutzim in Israel

98. moshavim in Israel

99. refugee camps in Australia

100. refugee camps in Austria

101. refugee camps in Italy

102. refugee camps in Jamaica

103. refugee camps in Sweden

104. refugee camps in Switzerland

105. refugee camps in the Netherlands

106. refugee camps in the United Kingdom

107. refugee camps in the United States

## B ChatGPT Prompt

For the ChatGPT baseline, for the first segment, we input the following prompt with the "system" role (filling in the actual list):

> "Here is a list of Holocaust-related event location categories:
> <Here comes the list of categories in appendix A>
> The user will give you a text segment and you will give back the location, out of the previous list, where the events in the segment took place. We emphasize that we want the event locations (from the list) and not the location of the interview. Give only the name of the category from the list with no additional text. If there is no location give the one that is the most probable.

Then we input the following with the "user" role (filling in the actual text):

> Segment: <Here comes the text segment>.

From the response, we extract a predicted location.

After a response was received we input the last query and response as part of the message history, in addition to the instructions. In cases where the response did not fit to any location in the list, we did not use it as part of the message history.

## C   Location Embedding Training

The training of location embeddings was done as NLI training with Multiple Negative Ranking loss. The first set of NLI examples was general-domain NLI data, which is a combination of the SNLI (Bowman et al., 2015) and MultiNLI (Williams et al., 2018). We trained with *learning-rate*$=2 \cdot 10^{-5}$.

We also obtained 2 sets of domain-specific examples. One set was generated from our annotated testimonies. For each segment that has an explicit location label, we generated two entailment examples of the form:

> <text segment> ==> "The event location is <label>"
> <text segment> ==> "The event location category is <label category>".

For each entailment example, we also used the opposite direction as entailment. Also, we created a contradiction example with a different location (or location category).

Another set was generated from the SF description for the locations and categories. For example, the location "Warsaw (Warsaw, Poland)" has the description:

> Coordinates:  52°15'N 21°00'E Capital of Poland 1929 Voivodship: Warsaw Jewish population in 1939: 393,950 1900-1917: Russia 1917-1918: Occupied by Germany 1918-1939: Poland 1939-1944: Occupied by Germany (Generalgouvernement) 1944-1945:  Liberated by Soviet troops 1945- : Poland.

For each of these descriptions, we generated an NLI example of the form:

> <description> ==> "The event location is <location>".

We used the opposite direction too and took a random other description as a contradiction.

We remark that although in our experiments we predicted the location categories, we still trained the vectors for detailed locations.

| Model | Edit | SM |
|---|---|---|
| IND. | 0.16 | 0.83 |
| CRF-G | 0.18 | 0.83 |
| CRF-L | 0.2 | 0.8 |
| HITRF | **0.11** | **0.89** |
| HITRF+EMB | 0.2 | 0.79 |

Table 2:  Performance for NBA biography team-tracking.

## D   Location Tracking in Biographies

As a validation, we implemented and evaluated our methods on another test domain.

**Data.**   We extracted from Wikipedia professional biographies of basketball players who played in the NBA. We applied our methods to the task of predicting the professional stage for each paragraph in the article.

Starting with the WikiBio dataset (Lebret et al., 2016), we selected all the biographies of basketball players who played in the NBA. We then obtained the full Wikipedia article (retrieved May 11, 2023) and filtered out the summary and sections that do not describe professional stages. We used the Wikipedia subsection titles as labels for each paragraph in the section.

We used labels for all current teams and legacy teams. We added labels for early careers, coaching and executive careers, and non-NBA teams. The final label set consists of 35 labels. For the full list see D.1.

The process yielded 655 biographies with an average of 16.9 labeled paragraphs. The average length is 1300 words.

**Training.**   We followed the same training procedure as with the Holocaust data. We changed to *learning-rate*$=10^{-6}$ for the segment-embedding transformer.

**Results and Discussion.**   For the NBA dataset, the location embeddings hurt the performance, even compared to the local classifier. In general, since the local classifier has such good performance, it proved difficult to add extra components without hurting the performance. Nevertheless, we still found that the use of hierarchical transformers yields a substantial improvement over the independent predictions. This shows that the global context is beneficial.

It's worth mentioning that we assume that the transformer models saw the NBA data with the headers during pretraining. This might be a reason for the strong performance of the local classification model. The LUKE model might have already learned useful embeddings, which outperform the ones we use here.

## D.1 List of NBA Biography Categories

1. Atlanta Hawks
2. Boston Celtics
3. Brooklyn Nets
4. Charlotte Hornets
5. Chicago Bulls
6. Cleveland Cavaliers
7. Coaching career
8. Dallas Mavericks
9. Denver Nuggets
10. Detroit Pistons
11. Early careers
12. Executive career
13. Golden State Warriors
14. Houston Rockets
15. Indiana Pacers
16. Los Angeles Clippers
17. Los Angeles Lakers
18. Memphis Grizzlies
19. Miami Heat
20. Milwaukee Bucks
21. Minnesota Timberwolves
22. New Orleans Pelicans
23. New York Knicks
24. Non-NBA team
25. Oklahoma City Thunder
26. Old team
27. Orlando Magic
28. Philadelphia 76ers
29. Phoenix Suns
30. Portland Trail Blazers
31. Retirement
32. Sacramento Kings
33. San Antonio Spurs
34. Toronto Raptors
35. Utah Jazz
36. Washington Wizards

## E Transition Matrices

We attach here (partial) heatmaps of the transition matrices obtained by the CRF models.

In figure 3 is a heatmap for the 35 most common location categories, obtained by the global CRF model. We can see some trends:

- For any category the most probable next step is to stay in the same category.

- After START the next common category is "concentration camps in poland". In general, after START, the common categories are in European countries.

Figures 4 and 5 are heatmaps for the 35 most common location categories, obtained by the local CRF model. Figure 4 is obtained for a segment describing a transport to a camp and figure 4 is obtained for a segment describing life in Tel Aviv after the war. We see that:

- The two matrices are very similar. This means that although the transitions depend on specific segments, it is still possible to extract some global trends.

- The second matrix has higher values in general. This might mean that this segment is interpreted as having more "mobility".

- As in the matrix for the global transitions, the highest values are for remaining in the same category. Also, there are similar patterns for transitions from START.

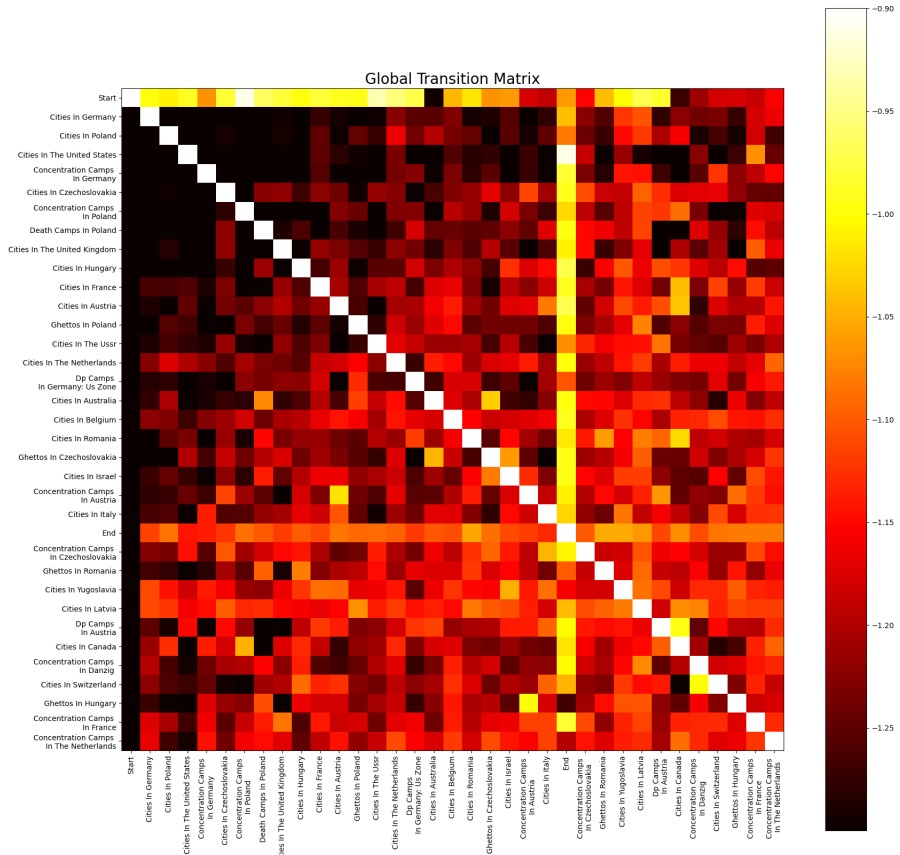

Figure 3: Heatmap of the global transition matrix for the global CRF model. We plotted only the 35 most frequent categories.

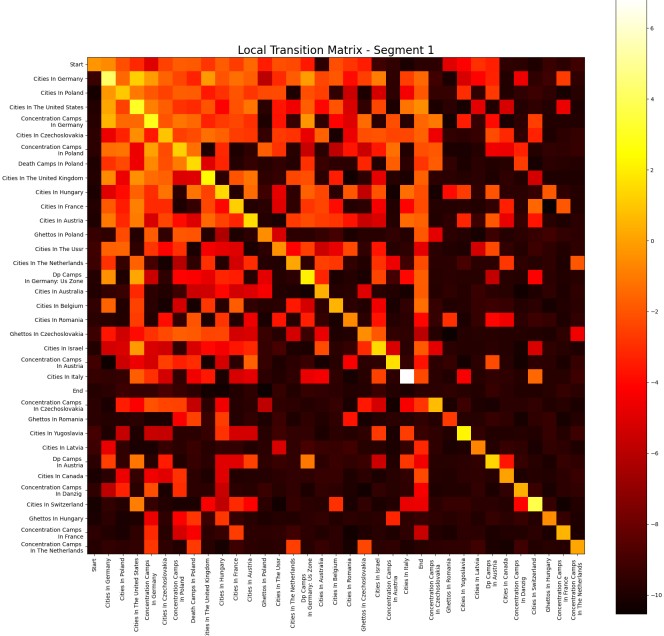

Figure 4: Heatmap of a local transition matrix for the local CRF model for a segment describing a transport to a camp. We plotted only the 35 most frequent categories.

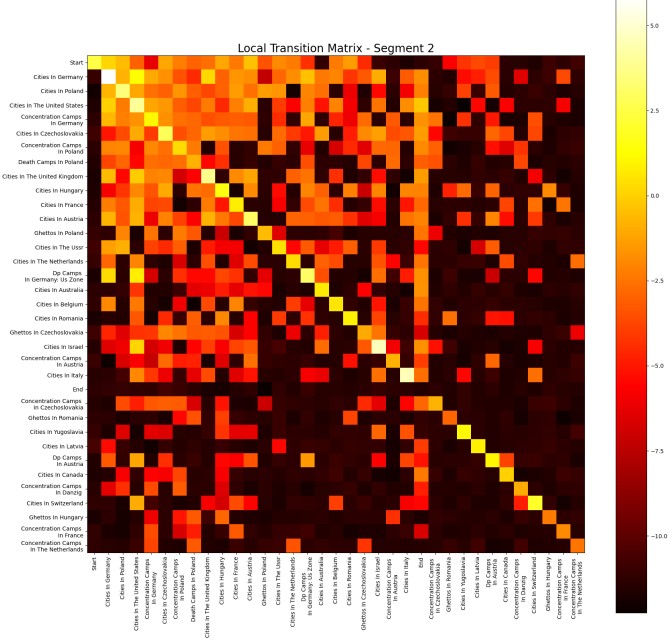

Figure 5: Heatmap of a local transition matrix for the local CRF model for a segment describing life in Tel Aviv after the war. We plotted only the 35 most frequent categories.