# OpenReview forum: "Event-Location Tracking in Narratives: A Case Study on Holocaust Testimonies"
_EMNLP/2023/Conference — EMNLP 2023 Main_

### Official Review · Reviewer_vwcJ · 2023-08-02

**Soundness:** 4

**Excitement:**

4: Strong: This paper deepens the understanding of some phenomenon or lowers the barriers to an existing research direction.

**Paper Topic And Main Contributions:**

The paper introduces a new task, narrative event-location trajectory extraction. The authors use a dataset of Holocaust survivor testimonies, which they convincingly argue is both extremely well-suited as a test bed for this new task (being a relatively large collection of topically-related narratives that take place in a small set of common locations) and a socially important domain in which to apply NLP techniques.

The authors encode the input narrative segments using a pretrained, transformer-based encoder and experiment with four sequence labeling approaches to predict the segment locations. Two approaches use a CRF on top of the encodings, one with a global location transition matrix learned from the entire training set, and one with a local transition matrix predicted for each segment from its encoding. The other two approaches use a second, hierarchical transformer encoder: one performs a standard multiclass probability prediction, and the other uses a pretrained location embedding matrix as the output layer.

The experimental results show that the proposed approaches outperform baseline methods that either do not use sequence information or perform local greedy prediction using the previous segment's predicted location. The authors also find that an LLM-based approach using ChatGPT did not perform well because it could not reliably produce predictions matching the given label set.

**Questions For The Authors:**

For the gold location sequences, is it the case that they always reflect the chronological order of locations in the story? I am wondering if locations can be given out of order, eg. "We had moved to City B from City A the year before," where City A is not mentioned in any other sentence. Section 5.1 mentions the special case of visits; were any other special cases observed in the dataset?

I am not sure that sequence-labeling metrics are the most appropriate to evaluate this task. For most of the segments, the label does not matter; the authors mention that, for most of the segments, the label is the same as that of the previous segment (and they use this heuristic to perform gold labeling). So if, for example, the gold location sequence is City A, City A, City A, City B, City B, then the second City A prediction seems to me less important than the first City B prediction. I would like to see some segmentation metrics added to the evaluation to focus on whether movement between locations is successfully predicted.
Update: This was addressed in the author rebuttal.

Do you have any intuition as to why the Greedy baseline underperforms Independent? I would have thought that access to some sequence information would be better than none, and it is not affected by the zero-shot baselines' formatting problems. I am also curious if you tried ensembling the Greedy baseline with a reversed version? Is global information required for good performance, or simply bidirectionality?

**Reasons To Accept:**

The proposed task is interesting and challenging. The authors motivate it well and chose a remarkably clean and well-suited dataset to demonstrate the task. The proposed approaches are sensible and perform well in the experiments.

**Reasons To Reject:**

The dataset used in the main experiments are not freely available (requiring permission from the Shoah Foundation archive), and the filtering step seems somewhat subjective, so the experimental results would be hard to reproduce.
Update: This second half of this concern was addressed in the author rebuttal.

While the proposed approaches are domain-agnostic, they do require there to be a fixed number of location labels, making them hard to apply to open-domain narratives.

**Reproducibility:**

3: Could reproduce the results with some difficulty. The settings of parameters are underspecified or subjectively determined; the training/evaluation data are not widely available.

**Reviewer Confidence:**

4: Quite sure. I tried to check the important points carefully. It's unlikely, though conceivable, that I missed something that should affect my ratings.

**Typos Grammar Style And Presentation Improvements:**

Typos:
- Section 5.1, line 450: , -> .

---

> ### Author Rebuttal · Authors · 2023-08-27
>
> Thank you for your appreciation of our work and suggestions. We will address some issues you raised and incorporate your suggestions.
>
> **Regarding the filtering of labels, you wrote (reasons to reject #1):**
> >The dataset used in the main experiments are not freely available (requiring permission from the Shoah Foundation archive), and the filtering step seems somewhat subjective, so the experimental results would be hard to reproduce. While the proposed approaches are domain-agnostic, they do require there to be a fixed number of location labels, making them hard to apply to open-domain narratives.
>
> The selection of location labels was based on the SF thesaurus hierarchy, which was created by Holocaust research experts. The thesaurus can be obtained for research purposes from the SF together with the testimonies, so this should not affect reproduction. It’s true that the methods require adaptation in order to generalize. We do, however, provide an additional dataset showing that generalization is possible.
>
> **Regarding unordered locations, you wrote (questions for authors #1):**
> > For the gold location sequences, is it the case that they always reflect the chronological order of locations in the story? I am wondering if locations can be given out of order
>
> The testimonies were conducted with general guidelines, and the order of events is mostly chronological, from childhood to post-war experiences. Nevertheless, the testimonies were not edited, and the stories can jump between events, times, and locations.
>
> **Regarding other metrics, you wrote (questions for authors #2):**
> > I am not sure that sequence-labeling metrics are the most appropriate to evaluate this task.
> >
> > I would like to see some segmentation metrics added to the evaluation to focus on whether movement between locations is successfully predicted.
>
> Since we also assume a correspondence between the text and locations (inducing a segmentation, for example), we do see importance in the exact sequence. We experimented with measures that are less sensitive to this (e.g., DTW or F1 score over changes only), but they were highly correlated to the sequence-evaluation methods and omitted for brevity. We will add them to the paper.
>
> **Regarding the greedy baseline’s performance and bidirectionality, you wrote (questions for authors #3):**
> > Do you have any intuition as to why the Greedy baseline underperforms Independent?
> >
> > I am also curious if you tried ensembling the Greedy baseline with a reversed version? Is global information required for good performance, or simply bidirectionality?
>
> As we mentioned in the discussion, it seems to be due to the delicate balance between the local signal and the global one (which includes the tendency to stay in the same place). For example, in cases where the location clearly changed but it’s not so obvious what the new location is, the greedy models showed a strong tendency to stay in the same place. This is in contrast to independent predictions which tended to change locations too often.
> We did not explore a bidirectional approach in the greedy baseline (although the CRF and HiTRF models are essentially bidirectional). Since we believe the issue is the tendency to stay in the same place, we believe that it won’t make a significant difference. We will check this empirically to verify.

---

### Official Review · Reviewer_VE7r · 2023-08-04

**Soundness:** 3

**Excitement:**

3: Ambivalent: It has merits (e.g., it reports state-of-the-art results, the idea is nice), but there are key weaknesses (e.g., it describes incremental work), and it can significantly benefit from another round of revision. However, I won't object to accepting it if my co-reviewers champion it.

**Paper Topic And Main Contributions:**

This paper proposes the task of event-location tracking in narrative texts. It proposes a set of methods to address this task (CRF variants, hierarchical transformers), and it focuses the evaluation on Holocaust survivor testimonies. It finds that the hierarchical transformer obtains the best results both on the main dataset and on an additional dataset discussed in the Appendix.

**Questions For The Authors:**

* L072-L076 and also L225-L231: can one can predict locations with NER and then filter out locations that are not part of the main event?
* L078-084: Can you please clarify the concept of trajectory? how the “post-war theme” is an event location?
* L102: “Larger context capabilities”: is there evidence to suggest that this is due to larger context or the nature of the task/dataset?
* L396: As mentioned in 479-480, BART-large was used as for zero-shot classification? Why not training this method instead of using a zero-shot classifier? Have you tried to use other models such as Flan-T5?
* L511: What form of gradient accumulation has been used?

**Reasons To Accept:**

* Interesting task of tracking locations in narratives.
* Interesting dataset on Holocaust survivor testimonies that could potentially also be used in other NLP tasks in the future (e.g., QA).

**Reasons To Reject:**

* The proposed CRF variants underperform the hiererchical transformers (HiTRF). This is the case both for the main dataset and for the dataset discussed in Appendix D.  Therefore the focus of the paper on modeling this task using a CRF (local and global context) should be down-graded. Moreover, location embeddings seem to provide additional information on the main dataset but not in the extra dataset (Appendix D). This may suggest that they can be useful for specific types of datasets only, but there is no discussion about this in the paper.
* More precise notation and details are needed in Sections 3 and 4 (see comments in Questions to Authors and Presentation improvements).
* The paper is lacking analysis on failure cases for the best model (HiTRF), and discussion on why certain cases fail. There’s some analysis in the Appendix but only for the CRF variants that underperform HiTRF.

**Reproducibility:**

4: Could mostly reproduce the results, but there may be some variation because of sample variance or minor variations in their interpretation of the protocol or method.

**Reviewer Confidence:**

4: Quite sure. I tried to check the important points carefully. It's unlikely, though conceivable, that I missed something that should affect my ratings.

**Typos Grammar Style And Presentation Improvements:**

* L212-L216: What is $x$? a sequence of paragraphs/sentences/tokens? does the predicted sequence of locations $y$ correspond to each element in $x$? Please clarify.
* L238-239: Please provide an example that motivates that implicitly mentioned locations
* L319: This has to be has to be more formal; please explain inputs and outputs, i.e., how tdo you agreggate transformer outputs (e.g., mean pooling)?
* L359: Please provide a reference to section 5.1 where more details are provided.
* L511: Please provide more details

---

> ### Author Rebuttal · Authors · 2023-08-27
>
> Thank you for your detailed review. We address here some of your concerns and we will incorporate many of your suggestions.
>
> **Regarding the results and their implications on structured prediction, you wrote (reasons to reject #1):**
> > The proposed CRF variants underperform the hiererchical transformers (HiTRF). This shows that, for this task, global context is more important than modeling sequential information, which in turn contradicts the focus of the paper on modeling dependencies as a structured prediction task.
>
> We are not sure we understand the point about the potential contradiction. By “structured prediction” we mean that the task should be addressed as predicting a whole structure and not independent local values. A common example of structured prediction is NER in IOB format, where an I label cannot come after a B label, so we cannot independently predict a label per token. Examples of non-structured prediction are most text classification tasks. In our case, the predicted structure is one sequence, corresponding to the whole document, and not a sequence of independent predictions for each segment. The reason for this is as we mentioned, that the sequence of locations exhibits dependencies between different parts of the document. By this definition, the decision is actually supported by our findings that models with larger contextual capabilities perform better.
>
> **Regarding the benefits of location embeddings for other domains, you wrote (reasons to reject #1):**
> > Moreover, location embeddings seem to provide additional information on the main dataset but not in the extra dataset (Appendix D). This may suggest that they can be useful for specific types of datasets only, but there is no discussion about this in the paper.
>
> This part seems to be a different concern. As we mention briefly in section 4.2, we believe that embeddings are beneficial for two reasons - (1) the ability to leverage the internal structure (like geographic proximity) of the labels, which is relevant for labels with descriptions, and (2) the ability to use the zero-shot capabilities of the models, which is mostly beneficial in cases of labels that are common in the unsupervised data but less in the supervised data. In the testimony data, the internal structure (1) follows the descriptions of the locations (from the SF Thesaurus), in which connections between locations are mentioned (e.g., for “Berlin”, the description includes its coordinates, its distance to Hamburg, and being part of Prussia). The zero-shot capabilities (2) were important because of the large number of possible locations (even if we measure prediction of the location categories, this is still based on mentions in the text, in which the number of mentioned locations is very large). In the Biography data, the local fine-tuned classifier showed very good performance (as described in Appendix D, results and discussion), so it seems that for the list of possible teams, which is not large, the zero-shot capabilities (2) are less crucial. It is true that the domain also seems significant, which is related to the internal structure (1). It seems that for the NBA team case, the geographic layout was less important compared to other factors, as geographical proximity does not seem like a significant factor in team switching. Other types of connections are possible but require explicit data which we did not have. We note that the biography domain mainly served as a proof of concept to demonstrate replicability across domains. Another consideration, which we mentioned in the discussion (in the appendix), is that the baseline method is strong, implying that the local signal is dominant. This means that adding the contextual signal is delicate, as we do not want to weaken the local one.
>
> **Regarding the analysis of failure cases of the HiTrf models, you wrote (reasons to reject #3):**
> > The paper is lacking analysis on failure cases for the best model (HiTRF), and discussion on why certain cases fail.
>
> We will add a discussion of this. Thank you for the suggestion.
>
> **Regarding the idea to filter out non-event locations, you wrote (questions for authors #1):**
> >  can one can predict locations with NER and then filter out locations that are not part of the main event?
>
> We actually tried similar approaches (like extracting all named locations and training a classifier based on this data). The problem is that identifying the location of the main event is hard. In addition, this approach failed miserably with segments that do not mention the main location (and sometimes do mention other locations). We note that even the locations that are relevant can be on different scales (city/country/district), or with different forms (requiring entity linking which is a problem in itself).
>
> **Regarding trajectories and the post-war theme, you wrote (questions for authors #2):**
> > Can you please clarify the concept of trajectory? how the “post-war theme” is an event location?
>
> The post-war theme is not an event location but rather a concept guiding the locations. What we try to convey in this example is the fact that migration from Europe to the US is very different from a post-war migration to a European country, in which the hometown usually plays a stronger role. Because migration to the US is a common post-war theme, it should not be viewed in the same way as migration to Europe, and both post-war migrations should not be viewed in the same way as other location changes before or during the war.
>
> **Regarding larger context capabilities, you wrote (questions for authors #3):**
> >  is there evidence to suggest that this is due to larger context or the nature of the task/dataset?
>
> In this paragraph, we just mention the correlation between context modeling abilities (which are inherent to the proposed architectures) and performance. Since the data is the same across the models, the most immediate potential conclusion is to ascribe this to context modeling capabilities. Of course, it is possible that our results are limited to specific properties of our domain, although we did try to address this point by experimenting in another domain.
>
> **Regarding the different models for zero-shot classification, you wrote (questions for authors #4):**
> >  BART-large was used as for zero-shot classification? Why not training this method instead of using a zero-shot classifier? Have you tried to use other models such as Flan-T5?
>
> We do not expect this to make much difference, since the main purpose of the zero-shot models is to leverage the connection between the mentioned locations and the label name. We interpret our findings as suggesting that the difficulty in the task is the implicitness of the locations (and the importance of context) and not the specific model’s identity. The same holds true for other similar-scale models. For this issue, training will be difficult, as we do not have many high-quality labeled segments with implicit-only locations. Also, in general, for closed-set classification with sufficient data, zero-shot methods (without large scale) are not much better than encoder-based ones.
>
> **Regarding the special form of gradient accumulation, you wrote (questions for authors #5):**
> > What form of gradient accumulation has been used?
>
> We will clarify this in the paper. What we do is the following: in the forward pass, one transformer ($E_{\theta_1}$) receives segments and outputs corresponding (pooled) encoding vectors, and a second transformer ($E_{\theta_2}$) receives all these vectors and outputs a sequence of locations. The loss is calculated by the outputs of  $E_{\theta_2}$, which are determined only after all segments were encoded (i.e., the input segments cannot be divided into batches). Encoding all segments before updating requires many copies of the computational graph, which requires a large amount of GPU memory. What we do is to run a forward pass for the $E_{\theta_1}$ without gradients. With these encodings we run a forward pass for $E_{\theta_2}$, with gradients, compute the loss, and backpropagate through $E_{\theta_2}$. With these (partial) gradients we run again through $E_{\theta_1}$, this time with gradients, but the previous calculations allow the use of batches. This whole procedure is purely for memory efficiency and does not affect the actual outputs and gradients.
>
> **Regarding the correspondence between the input and prediction sequences, you wrote (typos and improvements #1):**
> > What is $x$? a sequence of paragraphs/sentences/tokens? does the predicted sequence of locations $y$ correspond to each element in $x$?
>
> In the first paragraph, we describe the task in general, for any segmentation of the document into spans. The sequence corresponds element-wise (we will clarify this). Afterward, in L222 we mention that we use spans of multiple sentences, and in L425 we describe our data.
>
> **Regarding examples for implicitly mentioned locations, you wrote (typos and improvements #2):**
> > Please provide an example that motivates that implicitly mentioned locations
>
> We’ll add an example (here’s a simple one - when an event occurs over multiple segments, the location might be mentioned explicitly only in the first one).
>
> **Regarding pooling methods, you wrote (typos and improvements #3):**
> > please explain inputs and outputs, i.e., how tdo you agreggate transformer outputs?
>
> We used the default pooling module in the HuggingFace-transformers models, in the same manner that is used for sequence classification heads. Specifically, for sequence encoding we used LUKE, where the LukePooler applies a linear transformation and tanh activation to the embedding of a special token that is placed at the beginning of the sequence. We will clarify this in the paper.

---

### Official Review · Reviewer_adQB · 2023-08-10

**Soundness:** 4

**Excitement:**

3: Ambivalent: It has merits (e.g., it reports state-of-the-art results, the idea is nice), but there are key weaknesses (e.g., it describes incremental work), and it can significantly benefit from another round of revision. However, I won't object to accepting it if my co-reviewers champion it.

**Paper Topic And Main Contributions:**

 The paper introduces and explores the task of event-location tracking in narrative texts, proposes novel architectures with varying levels of context awareness, compares them against baselines, introduces methods for generating location embeddings, focuses on the Holocaust survivor testimonies as a case study, and validates the findings across different domains. The contributions of this paper lie in its formulation of a new task, the exploration of multiple architectural approaches, and the insights gained from analyzing the spatial dimension of narratives.

**Questions For The Authors:**

Question a: The motive to use category names (e.g., "Ghettos in Poland") instead of exact location names (e.g., "Warsaw (Warsaw, Poland: Ghetto)") should be clearly explained along with its potential implications if any

**Reasons To Accept:**

1. Architectural Exploration: The paper systematically explores various architectures for solving the event-location tracking task, ranging from global contextual models to narrow context baselines. This in-depth analysis provides valuable insights into the effectiveness of different approaches, aiding researchers and practitioners in selecting appropriate methods for similar tasks.

2. Location Embeddings: The introduction of methods for generating location embeddings, and the demonstration of their benefits over traditional string-based location representations, can have broader implications for representing geographical information within text data.

3. Ethical and Historical Relevance: The paper's focus on Holocaust survivor testimonies demonstrates a deep ethical and historical relevance. It showcases how NLP can contribute to the understanding and preservation of significant historical events and narratives.


**Reasons To Reject:**

1. Complexity vs. Performance Trade-off: The more advanced architectures may come with increased computational complexity. The paper should clearly discuss the trade-off between complexity and performance to provide practical insights for implementation.

2. Heuristic Labeling: The process of selecting location labels and handling segments without location labels involves heuristic approaches. This might introduce biases or errors in the labeling process, potentially affecting the quality of the training data.

3. Label Granularity: The decision to use category names (e.g., "Ghettos in Poland") instead of exact location names (e.g., "Warsaw (Warsaw, Poland: Ghetto)") might impact the precision of location tracking. The paper should discuss the implications of this choice and its potential effect on performance.

**Reproducibility:**

4: Could mostly reproduce the results, but there may be some variation because of sample variance or minor variations in their interpretation of the protocol or method.

**Reviewer Confidence:**

4: Quite sure. I tried to check the important points carefully. It's unlikely, though conceivable, that I missed something that should affect my ratings.

---

> ### Author Rebuttal · Authors · 2023-08-27
>
> Thank you for your review. We will address the issues you raised and incorporate your suggestions.
>
> **Regarding the Complexity vs. Performance Trade-off (reasons to reject #1):**
> > The paper should clearly discuss the trade-off between complexity and performance to provide practical insights for implementation.
>
> We will discuss this. In our experiments, the document-level models were significantly more complex than the baselines, but the performance was also significantly better. The time complexity of the deep CRF models and the HiTrf models was similar. The deep CRF ones were actually a bit more time-consuming, as they required more training epochs (as we reported in section 5.2). We did mention the complexity of the models regarding interpretability (end of section 7).
>
> **Regarding Heuristic Labeling, you wrote (reasons to reject #2):**
> > The process of selecting location labels and handling segments without location labels involves heuristic approaches. This might introduce biases or errors in the labeling process, potentially affecting the quality of the training data.
>
> The selection of location labels and categories was based on the SF thesaurus hierarchy, which was created by Holocaust research experts, so it was not arbitrary. The full thesaurus may be acquired from the SF together with the testimonies. As we mention in the paper, the annotation for segments without explicit labels was based on heuristics, and we realize that it introduces noise. For this reason, annotators manually corrected the test set so the heuristics’ effect on evaluation was minimal.
>
> **Regarding Label Granularity, you wrote (reasons to reject #3):**
> > The decision to use category names instead of exact location names might impact the precision of location tracking. The paper should discuss the implications of this choice and its potential effect on performance.
>
> The main reason for choosing the categories was to reduce the number of categories (from >1500 exact locations to ~100 categories). Reducing the number of categories inevitably results in some loss of information, but the categories still provide rich information useful for many tasks such as location-based segmentation, alignment between testimonies, and creation of high-level summaries. We will add a discussion about this.
>
> **Regarding the motive to use category names, you wrote (questions for authors a):**
> > The motive to use category names (e.g., "Ghettos in Poland") instead of exact location names (e.g., "Warsaw (Warsaw, Poland: Ghetto)") should be clearly explained along with its potential implications if any
>
> See response to reason to reject #3.

---

### Meta-Review · Area_Chair_xvcm · 2023-09-17

**Recommendation:** 2

**Metareview:**

This paper introduces the novel and socially impactful task of tracking event locations in personal narratives. The compelling application to Holocaust survivor testimonies underscores the moral importance of computationally analyzing such historical documents. The key technical contribution is a rigorous exploration of structured prediction architectures for extracting location sequences, including sequential conditional random fields (CRFs) and hierarchical transformers. Experiments demonstrate that modelling the global narrative context significantly improves prediction accuracy compared to greedy localized approaches.

The proposed task taps into a broader interest in narrative understanding and temporal commonsense reasoning. The testimony dataset appears remarkably well-suited, containing a large collection of topically-related personal accounts centered around a small set of canonical locations. However, transparency is needed around the subjective data filtering process to enable reproducibility without public access.

Reviewers praise the novelty, motivation, and dataset potential, but note a few areas for improvement. For example, Reviewer 2 believes the CRF focus should be downplayed given the superior performance of transformers. Reviewers 1 and 3 raise important concerns about aspects of the data creation process, such as the subjectivity of filtering location labels. Reviewer 3 offers multiple insightful suggestions like evaluating segmentation explicitly and analyzing bidirectionality. Overall the reviews indicate an intriguing task and technical approach while recommending improvements to evaluation, transparency, and framing.


Personally, I note the following: The CRF models underperform simple transformers, this suggests to me an overemphasis on sequential dependencies in framing and analysis. Additional segmentation metrics are recommended to better evaluate location trajectory extraction. Also, I am unconvinced about the results reported for ChatGPT. I have tried to replicate the task with ChatGPT and Claude and both appeared to have performed excellently on event location extraction. I concur that I have used different data for my experiments (mostly pseudo-text and not a real-life dataset like the one used in the paper). Also, I have used a simplified prompt than the example presented in the appendix of the paper. However, I do not think both factors should greatly impact the outcome. I think this calls for a more in-depth analysis on the part of the authors, especially to justify their approaches against out-of-the-box finetuning of LLMs for the same task. Also, an in-depth analysis of model failures could provide useful insights into remaining challenges. Lastly, details on hyperparameter tuning and model specifications would further support reproducibility.

---

### Decision · Program_Chairs · 2023-10-07

**Decision:**

Accept-Main

**Comment:**

This paper introduces the novel and socially impactful task of tracking event locations in personal narratives. The compelling application to Holocaust survivor testimonies underscores the moral importance of computationally analyzing such historical documents. The key technical contribution is a rigorous exploration of structured prediction architectures for extracting location sequences, including sequential conditional random fields (CRFs) and hierarchical transformers. Experiments demonstrate that modelling the global narrative context significantly improves prediction accuracy compared to greedy localized approaches.

The proposed task taps into a broader interest in narrative understanding and temporal commonsense reasoning. The testimony dataset appears remarkably well-suited, containing a large collection of topically-related personal accounts centered around a small set of canonical locations. However, transparency is needed around the subjective data filtering process to enable reproducibility without public access.

Reviewers praise the novelty, motivation, and dataset potential, but note a few areas for improvement. For example, Reviewer 2 believes the CRF focus should be downplayed given the superior performance of transformers. Reviewers 1 and 3 raise important concerns about aspects of the data creation process, such as the subjectivity of filtering location labels. Reviewer 3 offers multiple insightful suggestions like evaluating segmentation explicitly and analyzing bidirectionality. Overall the reviews indicate an intriguing task and technical approach while recommending improvements to evaluation, transparency, and framing.


Personally, I note the following: The CRF models underperform simple transformers, this suggests to me an overemphasis on sequential dependencies in framing and analysis. Additional segmentation metrics are recommended to better evaluate location trajectory extraction. Also, I am unconvinced about the results reported for ChatGPT. I have tried to replicate the task with ChatGPT and Claude and both appeared to have performed excellently on event location extraction. I concur that I have used different data for my experiments (mostly pseudo-text and not a real-life dataset like the one used in the paper). Also, I have used a simplified prompt than the example presented in the appendix of the paper. However, I do not think both factors should greatly impact the outcome. I think this calls for a more in-depth analysis on the part of the authors, especially to justify their approaches against out-of-the-box finetuning of LLMs for the same task. Also, an in-depth analysis of model failures could provide useful insights into remaining challenges. Lastly, details on hyperparameter tuning and model specifications would further support reproducibility.